

# Signatures of prescribed fire in the microbial communities of *Cornus florida* are largely undetectable five months post-fire

Beant Kapoor[1,*], Aaron Onufrak[1,*], William Klingeman III[2], Jennifer M. DeBruyn[3], Melissa A. Cregger[4], Emma Willcox[5], Robert Trigiano[1] and Denita Hadziabdic[1]

[1] Department of Entomology and Plant Pathology, University of Tennessee-Knoxville, Knoxville, Tennessee, United States
[2] Department of Plant Sciences, University of Tennessee-Knoxville, Knoxville, Tennessee, United States
[3] Department of Biosystems Engineering and Soil Science, University of Tennessee-Knoxville, Knoxville, Tennessee, United States
[4] Biosciences Division, Oak Ridge National Laboratory, Oak Ridge, Tennessee, United States
[5] Department of Forestry, Wildlife and Fisheries, University of Tennessee-Knoxville, Knoxville, Tennessee, United States
* These authors contributed equally to this work.

Corresponding author
Denita Hadziabdic,
dhadziab@utk.edu

## ABSTRACT

Prescribed burn is a management tool that influences the physical structure and composition of forest plant communities and their associated microorganisms. Plant-associated microorganisms aid in host plant disease tolerance and increase nutrient availability. The effects of prescribed burn on microorganisms associated with native ecologically and economically important tree species, such as *Cornus florida* L. (flowering dogwood), are not well understood, particularly in aboveground plant tissues (*e.g.*, leaf, stem, and bark tissues). The objective of this study was to use 16S rRNA gene and ITS2 region sequencing to evaluate changes in bacterial and fungal communities of five different flowering dogwood-associated niches (soil, roots, bark, stem, and leaves) five months following a prescribed burn treatment. The alpha- and beta-diversity of root bacterial/archaeal communities differed significantly between prescribed burn and unburned control-treated trees. In these bacterial/archaeal root communities, we also detected a significantly higher relative abundance of sequences identified as Acidothermaceae, a family of thermophilic bacteria. No significant differences were detected between prescribed burn-treated and unburned control trees in bulk soils or bark, stem, or leaf tissues. The findings of our study suggest that prescribed burn does not significantly alter the aboveground plant-associated microbial communities of flowering dogwood trees five months following the prescribed burn application. Further studies are required to better understand the short- and long-term effects of prescribed burns on the microbial communities of forest trees.

# INTRODUCTION

Globally, wildfires are growing in frequency and duration as a result of historical forest management practices and climate change (*Naficy et al., 2010*; *Zald & Dunn, 2018*; *McDowell et al., 2020*; *Tomshin & Solovyev, 2022*; *Weber & Yadav, 2020*; *Zhang et al., 2020*). Subsequently, forest resilience, or the ability of forests to recover from disturbances, such as wildfires, has been reduced and threatens forest sustainability (*McDowell et al., 2020*; *Stevens-Rumann et al., 2018*). Land managers often employ intentional and controlled application of prescribed fire to reduce fuel availability and the impacts of wildfires on forested ecosystems (*Ryan, Knapp & Varner Morgan, 2013*). Furthermore, prescribed fires can be used to reduce stand densities, maintaining species composition and structure of xeric forests, such as those present in the southeastern United States (USA) (*Schwartz et al., 2016*). In 2011 alone, approximately 6.4 million acres of land in the southeastern USA were subject to prescribed fire for forest management purposes (*Waldrop & Goodrick, 2012*). While prescribed fires are a valuable forest management tool, our understanding of their effects on the health of forest trees is limited, particularly in the context of the plant microbiome.

The plant microbiome, which consists of plant-associated microorganisms (*e.g.*, bacteria, fungi, protists, *etc.*), aids host plants in nutrient and water acquisition, stress tolerance, and plant defense (*Bell et al., 2019*). Despite the importance of the plant-associated microbial communities in supporting host plant health, research related to the responses of forest microbial communities to prescribed fire is often focused on the microorganisms of bulk and rhizosphere soils (*Borgogni et al., 2019*; *Hart et al., 2018*; *Qin & Liu, 2021*; *Rodríguez et al., 2018*). In general, fires, both wild and prescribed, reduce the richness of soil fungal communities, notably arbuscular mycorrhizal and ectomycorrhizal (EcM) fungi, which are key symbionts of plants, and aid in the uptake of phosphorus and nitrogen, respectively (*Dove & Hart, 2017*). Reductions in the richness of these and other fungi could have detrimental effects on host health by limiting their ability to obtain nutrients and increasing their susceptibility to pathogens (*Dowarah, Gill & Agarwala, 2021*). Soil bacterial communities also shift in response to wildfires and prescribed fires (*Brown et al., 2019*; *Dove et al., 2021*; *Hart et al., 2018*). Shifts in soil microbial communities post-fire can affect nutrient cycling, evidenced by changes in soil enzymatic activity related to the phosphorus, carbon, and nitrogen cycles (*Fairbanks et al., 2020*; *Rodríguez et al., 2018*). Furthermore, changes in soil microbial communities as a result of fire, impact which microorganisms colonize plant tissues, such as leaves and stems, highlighting the importance of characterizing plant tissue-associated microbial communities when assessing the effects of fire (*Dove et al., 2021*).

The effects of fire on microbial communities are context dependent, affected by factors such as the plant tissue examined, time between fire events and sampling dates, and fire intensity (*Barreiro & Díaz-Raviña, 2021*; *Dove et al., 2021*; *Dowarah, Gill & Agarwala,*

2021; *Qin & Liu, 2021*). At this time, our knowledge of the effects of prescribed fire on aboveground plant-associated microbial communities is severely limited. However previous research determined that aboveground plant-associated microbial communities are affected by prescribed fire, with shifts in leaf microbial community composition detected three months following prescribed burn treatment (*Dove et al., 2021*). It is currently unknown if effects of prescribed fire on aboveground tissues persist beyond three months, but responses of belowground microbial communities to prescribed fire are highly temporally variable highlighting a need to assess the responses of aboveground microbial communities to prescribed fires at different timescales (*Dove & Hart, 2017*). Furthermore, the composition of leaf microbial communities was observed to shift along a fire intensity gradient (*Dove et al., 2021*). This is of interest because the intensity of prescribed fires differs from wildfires, and as such, the effects of prescribed fires on plant-associated microbial communities will likely differ from those of wildfires. Wildfires typically burn hotter for longer times compared to prescribed burns and are therefore more intense, depleting more of the aboveground biomass and resulting in greater nutrient volatilization (*Oliver, Callaham & Jumpponen, 2015*). Thus, there is a need to understand how prescribed fires affect the microbial communities associated with plant tissues.

To determine the effects of prescribed fires on plant-associated microbial communities, the objective of our study was to characterize changes in the diversity and compositions of fungal and bacterial/archaeal communities across five different plant-associated niches of *Cornus florida* L. (flowering dogwood) trees, five months following the application of a prescribed burn. Flowering dogwood is a deciduous understory tree species native to the eastern USA (*Sharma et al., 2005*; *Harrar & Harrar, 1962*; *Sork et al., 2005*). Flowering dogwood fruits are an important source of nutrition for forest wildlife as they contain some of the highest levels of calcium and fat available among forest plant resources (*Halls & Epps, 1969*). In addition to being an important forest resource, flowering dogwood is an economically important nursery crop contributing over $31 million in revenue to USA annually (*United States Department of Agriculture–National Agricultural Statistics Service (USDA-NASS), 2017*). Most notably, flowering dogwood trees are present within the Great Smoky Mountain National Park (North Carolina and Tennessee, USA) an area in which prescribed fires are employed to reduce stand density to maintain the composition of the pine-oak forests in this region (*Schwartz et al., 2016*). At this time, the effects of prescribed fire on the health of flowering dogwood trees have been poorly characterized. However, the density of flowering dogwoods in the Great Smoky Mountain National Park was higher in plots burned by a wildfire, likely as a product of reduced stand density (*Jenkins & White, 2002*). This suggests that prescribed burns have positive effects on the growth of flowering dogwood trees (*Jenkins & White, 2002*).

To accomplish our objective, soil, roots, bark, stem, and leaf samples were collected from flowering dogwood trees in unburned (control) and prescribed burn plots five months following burn application. Microbial communities were characterized using amplicon sequencing to analyze changes in bacterial/archaeal and fungal diversity and community composition in response to prescribed burn. We hypothesized that for all

niches (*i.e.*, bulk soils, roots, bark, stems, and leaves), microbial diversity would decrease and community composition would be significantly altered in response to the prescribed burn treatment. We further hypothesized that the microbial diversity of belowground niches (*i.e.*, bulk soils and roots) would be impacted more severely by fire compared to aboveground niches (*i.e.*, bark, stems, and leaves) because of exposure to higher temperatures resulting from closer proximity to fire. Lastly, we expected a greater prevalence of pyrophilic or fire-loving microorganisms in the prescribed burn plots.

## MATERIALS AND METHODS

### Site description and study design

This study was conducted at the University of Tennessee (UT) Highland Rim Forest Unit Research and Education Center in Tullahoma, TN, USA (35.32, −86.15). The experimental site was divided into two plots, each approximately 8,000 m$^2$ in area, containing established flowering dogwood trees. One plot was randomly assigned to the prescribed burn treatment, whereas the other was chosen as the unburned control plot. Within each plot, 10 trees were selected randomly (n = 20; average diameter at breast height 7.7 ± 1.0 cm, average height 8.0 ± 1.5 m, average age 38.5 ± 4.9 years; Table S1) for post-burn sample collection. The sampled flowering dogwood trees were present in a mixed forest stand which included *Quercus alba*, *Q. lyrata*, *Diospyros virginiana*, and *Pinus strobus*. There were on average four trees within a 5 m radius of sampled flowering dogwood trees. Soils in each plot were a Baxter cherty silt loam (Web Soil Survey, http://websoilsurvey.sc. egov.usda.gov/; accessed February 6th, 2023). Data were collected as previously described by *Kapoor (2020)*.

### Burn application

The prescribed burn was applied on 28 March 2019 by certified personnel from the UT Forest Resources Research & Education Center staff (Oak Ridge, TN, USA) and Tennessee Division of Forestry employees. Prescribed burn has not been applied on this site prior to this study. Litter present on the east side of the plot was ignited at 1:45 pm (CDT) (Fig. S1). The burn progressed westward as head-fire however shifted to back-fire at around 1:55 pm (Fig S1). Litter on the west side of the plot was ignited at 2:25 pm. The site burned for approximately 1 h and naturally extinguished upon depletion of leaf litter (Fig. S1D). Temperature was recorded by an infra-red imaging camera at the soil-air interface for four of the ten flowering dogwood trees under the prescribed burn treatment (Fig. S2). The average temperature of the burn was 210 ± 133°C (max observed temperature 788.5 °C). The average air temperature was 18.3 ± 1.7 °C with average humidity of 33.8 ± 0.7%. The average wind speed was 9.84 ± 2.62 mph.

### Soil and plant sample collection

To determine the short-term effects of prescribed fire on flowering dogwood microbial communities, samples were collected from ten trees per treatment (*n* = 20; Table S1). Samples were collected from bulk soils, roots, bark, stems, and leaves on 27–28 August 2019 (Fall), 152 days (approximately five months) after the burn application. Collection

tools were cleaned and surface sterilized with 70% ethanol in between niches and trees to prevent cross-contamination of samples. For collection of bulk soils and roots, plant debris was removed from the base of each tree. Bulk soils were then collected using a sterilized stainless-steel soil probe (15 cm deep × 3 cm diameter). Soil was collected from the four cardinal directions, approximately 0.3 m from the base of each tree. Plant debris and larger roots were removed by hand, and the soil cores were pooled per tree and homogenized in the field. For microbial analyses, approximately 5 g of soil was subsampled from the homogenized bulk soil sample and immediately placed into liquid nitrogen for transport to the lab. Samples were stored at −80 °C until DNA extraction. The remaining bulk soil was stored at 4 °C for analyses of soil physicochemical properties. For measurement of soil physicochemical properties, soil was air-dried, ground with a mortar and pestle, passed through a 2 mm sieve, and sent to Brookside Laboratories (New Bremen, OH, USA) for analyses of pH (1:1; *McLean, 1982*), soil organic matter (SOM; loss on ignition 360 °C; *Schulte & Hopkins, 1996*), ammonium ($NH_4$; *Dahnke, 1990*), and Mehlich III extractable boron (B), calcium (Ca), copper (Cu), iron (Fe), potassium (K), magnesium (Mg), manganese (Mn), phosphorus (P), sulfur (S), and zinc (Zn) (*Mehlich, 1984*).

Lateral roots were traced from the base of each study tree and roots up to approximately 5–10 cm in depth and 1 mm in diameter were collected. Bark samples were obtained by shaving a small section of exposed cambium approximately 30 cm above the base of the tree. A pole pruner was used to cut 3–4 branches with leaves from the four cardinal directions from each tree. For stem samples, approximately 2–3 8 cm in length by 2 mm in diameter branch sections were cut from each sampled branch and pooled per tree for 10–12 bark sections per tree. For leaf sample collection, 3–4 mature leaves were collected from the aforementioned branch sections for a total of 10–12 leaves per tree. Collected root, bark, stem, and leaf tissues were bulked per niche per tree (20 samples per niche) and immediately placed in liquid nitrogen and transported to the lab where they were stored at −80 °C until DNA extraction.

## DNA extraction and library preparation

DNA was extracted from root, bark, stem, and leaf tissues following the E.Z.N.A. Plant DNA DS Mini Kit protocol (Omega Bio-tek, Norcross, GA, USA). For DNA extraction from soil, the PowerLyzer® PowerSoil® DNA Isolation Kit protocol (Qiagen, Carlsbad, CA, USA) was followed. Extracted DNA was stored at −20 °C until PCR, library preparation, and sequencing on the Illumina MiSeq platform.

For library preparation, DNA extracts were sent to Psomagen, Inc. (Rockville, MD, USA). A two-step PCR approach was used to barcode templates with the following modifications. To amplify the 16S rRNA V3–V4 regions, the 341F forward and 805R reverse primer pair were used (*Herlemann et al., 2011*). To improve taxonomic coverage of fungal taxa, the ITS2 region was amplified using six ITS3 forward and two ITS4 reverse primers (*Martin & Rygiewicz, 2005*; *Cregger et al., 2018*). For both the 16S and ITS amplifications, primary PCR was performed in 25 µL reactions containing 2× KAPA HiFi hot start ready mix, 2.5 µL of genomic DNA, and 2 µM of forward and reverse primers each. Thermocycler conditions for the primary PCR were initial denaturation at 95 °C for

30 s, followed by 30 cycles of denaturation at 95 °C for 30 s, annealing at 55 °C for 30 s, and elongation at 72 °C for 30 s, and then final elongation step at 72 °C for 5 min. Primary PCR products were cleaned with 20 μL of AMPure beads and eluted in 50 μL of hydroxymethyl-aminomethane (TRIS) buffer. Secondary PCR had purified DNA tagged with barcoded forward and reverse indices in the 50 μL reaction having 5 μL of genomic DNA. Thermocycler conditions for secondary PCR were initial denaturation at 95 °C for 3 min, followed by 8 cycles of denaturation at 95 °C for 30 s, annealing at 55 °C for 30 s, elongation at 72 °C for 30 s and then a final elongation step at 72 °C for 5 min. The product was quantified on a NanoDrop 1,000 spectrophotometer (NanoDrop Products, Wilmington, DE, USA). After the second PCR, the samples were pooled based on the Bioanalyzer (Agilent, Santa Clara, CA, USA) reading. The final loading concentration of the pooled samples was 4 pM. Illumina MiSeq sequencing was carried out using a 20% PhiX spike on a V2, 500 cycle flow cell reading 2 × 250 bp. Raw amplicon sequences are located under the NCBI SRA BioProject PRJNA754133.

## Bacterial (16S) and fungal ITS sequence processing

The resulting V3–V4 and ITS2 region reads were processed into amplicon sequence variants (ASVs) following the DADA2 16S rRNA gene (https://benjjneb.github.io/dada2/tutorial.html) and ITS2 region (https://benjjneb.github.io/dada2/ITS_workflow.html) workflows in R version 4.1.0 (*Callahan et al., 2016*; *R Development Core Team, 2011*). Using cutadapt, primers were removed before quality filtering and denoising for both V3–V4 and ITS2 reads (*Martin, 2011*). Prior to denoising with DADA2, sequences were quality filtered using the *filterAndTrim* command with the maximum expected errors parameter set to two for both forward and reverse reads (maxEE = 2,2). Chimeric sequences were removed using the *removeBimeraDenovo* function using the consensus method. The naïve-bayes classifier trained on the SILVA r.138 database for V3–V4 merged sequences and the UNITE database v.8.3 for ITS2 merged sequences was used to assign taxonomy to the resulting ASVs, for bacteria/archaea and fungi, respectively (*Nilsson et al., 2019*; *Quast et al., 2013*). Following taxonomic assignment, V3–V4 sequences not identified to a bacterial/archaeal phylum or those identified as chloroplast or mitochondria were removed. Additionally, ITS2 sequences not assigned to a fungal phylum were removed from the dataset. Codes used to analyze 16S and ITS sequence data can be accessed *via* https://doi.org/10.5281/zenodo.7948602.

## Alpha- and beta-diversity analyses

All statistical analyses were completed in R version 4.1.0 (*R Development Core Team, 2011*). To test for significant differences in soil physicochemical properties between unburned control and prescribed burn plots, two-sample t-tests were performed for all of the aforementioned soil physicochemical properties using the *t.test* function from the *stats* package (*R Development Core Team, 2011*). Data were log-transformed for variables that did not meet the assumptions of normality and homoscedasticity, tested using the Shapiro-Wilk's test of normality and the Levene's test for homogeneity of variance, respectively. A principal component analysis (PCA) with the *prcomp* function from the *stats* package (*R*

*Development Core Team, 2011*) was used to examine how soil physicochemical properties differed between unburned control and prescribed burn plots in multivariate space.

To control for potential biases caused by different DNA extraction methods, statistical analyses for the bulk soil communities and plant-associated communities were performed separately (*Lim et al., 2018*). Prior to computing alpha- and beta-diversity metrics, microbial communities were rarefied using the *rrarefy* function from the *vegan* package to account for differences in sequencing depth among samples (*Oksanen et al., 2023*; *Weiss et al., 2017*). Rarefaction cut-offs were chosen by reviewing rarefaction curves constructed using the *rarecurve* function from the *vegan* package and selecting a value that would maximize per sample sequencing depth and minimize sample loss (*Oksanen et al., 2023*). To determine how the prescribed burn affected alpha-diversity of flowering dogwood-associated and bulk soil microbial communities, Hill numbers were calculated (q0 (observed ASV richness), q1 (Shannon diversity), and q2 (Inverse Simpson)) using the *hill_taxa* function from the *hillR* package (*Li, 2018*). For flowering dogwood-associated microbial communities, a type III, two-way analysis of variance (ANOVA) was used to test for significant differences in alpha-diversity by plant niche, prescribed burn treatment, and their interaction using the *aov* and *Anova* functions from the *stats* and *car* packages, respectively (*R Development Core Team, 2011*; *Fox & Weisberg, 2018*). If the interaction term was not significant, then the interaction term was dropped from the model and only the additive effects of plant niche and prescribed burn treatment were included in the model. Hill numbers (q0–q2) were log-transformed to meet the assumptions of normality and homoscedasticity. If a significant effect was detected, a Tukey's *post-hoc* test was used to identify between group differences using the *TukeyHSD* function from the *stats* package (*R Development Core Team, 2011*). For bulk soils, we used a two-sample *t*-test as described above to test for significant differences in alpha-diversity between unburned control and prescribed burn treatments.

To characterize the responses of bulk soil and flowering dogwood-associated microbial communities to prescribed burn treatment, we calculated the relative abundances of the rarefied ASV count data and computed sample-wise Bray-Curtis distances using the *vegdist* function in *vegan* (*Oksanen et al., 2023*). Distances were visualized with a principal coordinate analysis (PCoA) using the *pcoa* function in the *ape* package (*Paradis & Schliep, 2019*). The *betadisper* function from *vegan* was used to test for multivariate homogeneity of group dispersions (*Oksanen et al., 2023*). To test for differences in flowering dogwood-associated microbial communities based on plant niche, prescribed burn treatment, and their interaction, a permutational multivariate analysis of variance (PERMANOVA) with 999 permutations using the *adonis* function from *vegan* was used (*Oksanen et al., 2023*). A PERMANOVA was also used to test for differences in community composition of bulk soil microbial communities based on prescribed burn treatment.

For any belowground microbial communities (*i.e.*, bulk soil and roots) that significantly differed by prescribed burn treatment, we characterized the relationships between soil physicochemical properties and microbial community composition by performing a distance-based redundancy analysis (dbRDA) using the *dbrda* function from *vegan* (*Oksanen et al., 2023*). The significance of model terms was determined using the *anova.*

*cca* function from *vegan*. To reduce the effects of multicollinearity, the scope of the final model was limited to include variables with variance inflation factors (VIFs) less than or equal to five. VIFs were calculated using the *vif* function from *car* package (*Fox & Weisberg, 2018*).

### Differential abundance testing

For microbial communities that significantly differed by burn treatment, we used DESeq2 on unrarefied ASV tables to identify differentially abundant ASVs between unburned control and prescribed burn treatments (*Weiss et al., 2017*). Due to the presence of zero counts in each ASV, a pseudo-count of one was added to all ASV counts to allow for the computation of log geometric means (*Olivas-Martínez et al., 2022*). Due to increased false discovery rates when using DESeq2 with small sample sizes and large differences in sequencing depth between samples, we chose to interpret DESeq2 results conservatively, using them as a guide to identify families or genera to target for differential abundance testing using traditional parametric methods (*Weiss et al., 2017*). To test for significant differences in the relative abundances of candidate families and genera identified with DESeq2, two sample t-tests were used as described previously.

## RESULTS

### Bacterial (16S) and fungal (ITS) sequence processing

A total of 10.7 million V3–V4 paired-end raw reads were processed into 1.5 million sequences across 16,624 bacterial/archaeal ASVs. For the ITS2, 13.1 million paired-end raw reads were processed into 2.4 million sequences across 4,903 fungal ASVs. Prior to alpha- and beta-diversity analyses, bacterial/archaeal and fungal communities were rarefied to account for differences in sequencing depth between samples and allow for between and within niche comparisons (Table S2; Figs. S3 and S4). Rarefaction cut-offs were chosen to minimize sample loss and maximize per sample sequencing depth.

### Soil physicochemical properties

Soil K and $NH_4$ significantly differed between the unburned control and prescribed burn-treated plots, with higher concentrations of K and $NH_4$ in the unburned control plot (Table 1; Fig. S5). TEC, soil pH, SOM, P, Ca, Mg, K, Fe, Cu, and Zn did not significantly differ between unburned control and prescribed burn treated plots (Table 1). The first two principal components explained 55.2% of the variation in soil physicochemical properties (Fig. S6). Soil samples were primarily separated by prescribed burn treatment along PC2, which was composed predominantly of pH, Mn, TEC, $NH_4$, K, and Ca (Fig. S6). Soil samples separated within prescribed burn treatments along PC1, which was comprised primarily of P, Mg, Ca, Cu, and TEC (Fig. S6).

### Alpha-diversity analyses

Hill numbers (q0–q2) significantly differed by niche ($P < 0.05$) but not by prescribed burn treatment or their interaction for flowering dogwood-associated fungal communities and bulk soil fungal and bacterial/archaeal communities (Tables 2 and 3; Figs. 1A–1C and 2).

**Table 1 Mean and standard deviation of soil physicochemical properties of bulk soils collected from the bases of unburned control and prescribed burn treated flowering dogwood (*Cornus florida*) trees with summary statistics of two sample *t*-tests.**

| Soil variable | Treatment | | t-test statistic | | |
| --- | --- | --- | --- | --- | --- |
| | Control | Prescribed burn | df | t | P |
| TEC (meq/100 g) | 2.9 ± 1.2 | 3.0 ± 0.7 | 18 | 0.1 | 0.9 |
| pH | 4.7 ± 0.3 | 4.6 ± 0.2 | 18 | −1.3 | 0.2 |
| SOM (%) | 2.6 ± 0.5 | 2.6 ± 0.5 | 18 | 0.2 | 0.9 |
| S (ppm) † | 15.1 ± 1.9 | 15.1 ± 1.7 | 18 | −0.003 | 1.0 |
| P (mg/kg) | 6.8 ± 1.9 | 7.2 ± 1.4 | 18 | 0.5 | 0.6 |
| Ca (mg/kg) | 150.9 ± 65.9 | 156.0 ± 59.7 | 18 | 0.2 | 0.9 |
| Mg (mg/kg) | 25.8 ± 5.2 | 24.4 ± 3.5 | 18 | −0.7 | 0.5 |
| **K (mg/kg)** | **42.5 ± 7.1** | **35.2 ± 6.3** | **18** | **−2.4** | **0.03** |
| Fe (mg/kg) | 83.1 ± 9.7 | 89.9 ± 12.4 | 18 | 1.4 | 0.2 |
| Mn (mg/kg) † | 39.0 ± 24.7 | 22.9 ± 14.1 | 18 | −0.18 | 0.09 |
| Cu (mg/kg) | 0.8 ± 0.1 | 0.8 ± 0.05 | 18 | 0.09 | 0.9 |
| Zn (mg/kg) | 0.9 ± 0.2 | 0.9 ± 0.3 | 18 | 0.5 | 0.6 |
| **$NH_4$ (ppm)** | **17.4 ± 3.2** | **13.1 ± 2.6** | **18** | **−3.3** | **0.004** |

Note:
Text in bold represents properties that significantly differed by prescribed burn treatment ($P < 0.05$). The † indicates data that was log transformed to meet parametric assumptions of t-test.

**Table 2 Summary statistics for two-way analysis of variance (ANOVA) and two sample *t*-tests of Hill numbers (q0–q2) from flowering dogwood (*Cornus florida*) associated fungal and bacterial/archaeal communities by niche, prescribed burn treatment, and the interaction of plant niche and prescribed burn treatment.**

| Community | Niches | q0 (ASV richness) | | | | | | | | | | | |
| --- | --- | --- | --- | --- | --- | --- | --- | --- | --- | --- | --- | --- | --- |
| | | Intercept | | | Plant niche | | | Burn treatment | | | Plant niche * Burn treatment | | |
| | | df | F | P | df | F | P | df | F | P | df | F | P |
| Fungi | All plant† | 1.69 | 819.11 | <0.0001 | 3.69 | 14.14 | <0.0001 | 1.69 | 0.68 | 0.41 | 3.66 | 0.63 | 0.60 |
| | | | | | | | | df | t | P | | | |
| | Bulk soil | – | – | – | – | – | – | 14.58 | 1.31 | 0.21 | – | – | – |
| Bacteria/Archaea | All plant† | 1.67 | 354.22 | <0.0001 | 3.67 | 30.96 | <0.0001 | 1.67 | 0.62 | 0.43 | **3.67** | **3.69** | **0.02** |
| | | | | | | | | df | t | P | | | |
| | Bulk soil | – | – | – | – | – | – | 14.94 | 0.05 | 0.96 | – | – | – |
| | | q1 (Shannon Entropy) | | | | | | | | | | | |
| | | Intercept | | | Plant niche | | | Burn treatment | | | Plant niche * Burn treatment | | |
| Fungi | All plant† | 1.69 | 195.82 | <0.0001 | 3.69 | 10.99 | <0.0001 | 1.69 | 1.18 | 0.28 | 3.66 | 0.86 | 0.47 |
| | | | | | | | | df | t | P | | | |
| | Bulk soil | – | – | – | – | – | – | 17.68 | 1.53 | 0.14 | – | – | – |
| Bacteria/Archaea | All plant† | 1.67 | 2,077.86 | <0.0001 | 3.69 | 31.39 | <0.0001 | 1.67 | 0.62 | 0.43 | **3.67** | **5.521** | **0.002** |
| | | | | | | | | df | t | P | | | |
| | Bulk soil | – | – | – | – | – | – | 14.86 | 0.17 | 0.86 | – | – | – |
| | | q2 (Inverse Simpson) | | | | | | | | | | | |
| | | Intercept | | | Plant niche | | | Burn treatment | | | Plant niche * Burn treatment | | |

(Continued)

| | | q0 (ASV richness) | | | | | | | | | | | | |
|---|---|---|---|---|---|---|---|---|---|---|---|---|---|---|
| | | Intercept | | | Plant niche | | | Burn treatment | | | Plant niche * Burn treatment | | | |
| Community | Niches | df | F | P | df | F | P | df | F | P | df | F | P | |
| Fungi | All plant† | 1.69 | 135.47 | <0.0001 | 3.69 | 11.26 | <0.0001 | 1.69 | 0.90 | 0.35 | 3.66 | 0.53 | 0.67 | |
| | | | | | | | | df | t | P | | | | |
| | Bulk soil | – | – | – | – | – | – | 15.98 | 1.13 | 0.28 | – | – | – | |
| Bacteria/Archaea | All plant† | 1.67 | 1,061.74 | <0.0001 | 3.67 | 19.69 | <0.0001 | 1.67 | 0.32 | 0.57 | 3.67 | 3.30 | 0.03 | |
| | | | | | | | | df | t | P | | | | |
| | Bulk soil | – | – | – | – | – | – | 15.39 | 0.33 | 0.74 | – | – | – | |

**Note:**
Text in bold represents Hill numbers that significantly differed by prescribed burn treatment ($P < 0.05$). The † indicates data that was log transformed to meet assumptions of ANOVA test.

Fungal communities associated with aboveground tissues (*i.e.* bark, leaves, and stems) had higher alpha-diversity than those associated with roots except when measured using the Inverse Simpson index (q2), where root and leaf communities had similar levels of alpha-diversity (Tables 2 and 3; Figs. 1A–1C). For flowering dogwood-associated bacterial communities, Hill numbers (q0–q2) differed significantly ($P < 0.05$) by the interaction between burn treatment and plant niche (Tables 2 and S3; Figs. 1D and 1E). Hill numbers (q0–q2) of prescribed burn bacterial/archaeal root communities were significantly lower compared to unburned control roots (Tables 1 and S3; Figs. 1D and 1E).

## Beta-diversity analyses

Niche explained 18% of the variation in plant-associated fungal communities and 27% of the variation in plant-associated bacterial/archaeal communities ($P < 0.05$; Table 4). The composition of flowering dogwood-associated fungal and bacterial/archaeal communities did not significantly differ by burn treatment or the interaction of niche and burn treatment ($P > 0.05$; Table 4; Figs. 3A and 4A). However, due to significant differences in within group variability (heteroscedasticity) when all plant niches were considered together, beta-diversity analyses were also conducted on niches individually. Within group variation was homoscedastic, or had equal within group variability at the burn treatment level, when niches were considered individually. Fungal community composition was not significantly different by burn treatment for all individual plant niches ($P > 0.05$; Fig. 3). The composition of root-associated bacterial/archaeal communities did significantly differ by prescribed burn treatment ($P < 0.05$; Table 4; Fig. 4B). Prescribed burn treatment explained 8.3% of the variation in community composition of root-associated bacterial/archaeal communities (Table 4; Fig. 4B). The burn treatment did not significantly affect the composition of any other flowering dogwood associated bacterial/archaeal communities (Table 4; Fig. 4C–4E). Furthermore, bulk soil fungal and bacterial/archaeal community compositions were not significantly affected by the burn treatment ($P > 0.05$; Fig. 5).

**Table 3 Results of *post-hoc* Tukey's tests calculated for analysis of variance (ANOVA) of Hill numbers (q0–q2) from plant-associated fungal communities of flowering dogwood (*Cornus florida*) trees.**

| | | 0D (ASV richness) | | | |
|---|---|---|---|---|---|
| | | **Roots** | **Bark** | **Stem** | **Leaves** |
| | | **P** | **P** | **P** | **P** |
| Fungi | Roots | – | **0.0001** | **0.0006** | **<0.0001** |
| | Bark | – | – | 0.96 | 0.23 |
| | Stem | – | – | – | 0.10 |
| | Leaves | – | – | – | – |
| | | 1D (Shannon Entropy) | | | |
| | | Roots | Bark | Stem | Leaves |
| | | P | P | P | P |
| Fungi | Roots | – | **0.0009** | **<0.0001** | **0.006** |
| | Bark | – | – | 0.29 | 0.92 |
| | Stem | – | – | – | 0.09 |
| | Leaves | – | – | – | – |
| | | 2D (Inverse Simpson) | | | |
| | | Roots | Bark | Stem | Leaves |
| | | P | P | P | P |
| Fungi | Roots | – | **0.004** | **<0.0001** | 0.26 |
| | Bark | – | – | 0.16 | 0.28 |
| | Stem | – | – | – | **0.001** |
| | Leaves | – | – | – | – |

**Note:**
Text in bold represents Hill numbers that significantly differed by prescribed burn treatment ($P < 0.05$). Data was log transformed to meet assumptions of ANOVA and post-hoc Tukey's test.

Soil physicochemical properties explained 48.9% of the variation in root bacterial/archaeal communities. The concentration of $NH_4$ was a significant driver of differences in root bacterial/archaeal communities ($P < 0.05$; Fig. 6). Lower concentrations of $NH_4$ were associated with the bacterial/archaeal root communities of prescribed burn-treated trees (Fig. 6). The remaining soil physicochemical properties analyzed did not significantly explain differences in community composition ($P > 0.05$), but the concentrations of K, Mg, and pH did help differentiate between unburned control and prescribed burn treated trees. The root bacterial/archaeal communities of prescribed burn trees were associated with lower concentrations of K and Mg and higher pH values compared to unburned control trees.

## Community composition and differential abundance tests

In root-associated fungal communities of unburned control flowering dogwood trees, the majority of sequences were assigned to the phylum Ascomycota and in prescribed burn trees the majority were assigned to the phylum Basidiomycota (Fig. 7A). In the remaining flowering dogwood-associated fungal communities, Ascomycota sequences comprised the greatest proportion of total sequences for both unburned control and prescribed burn

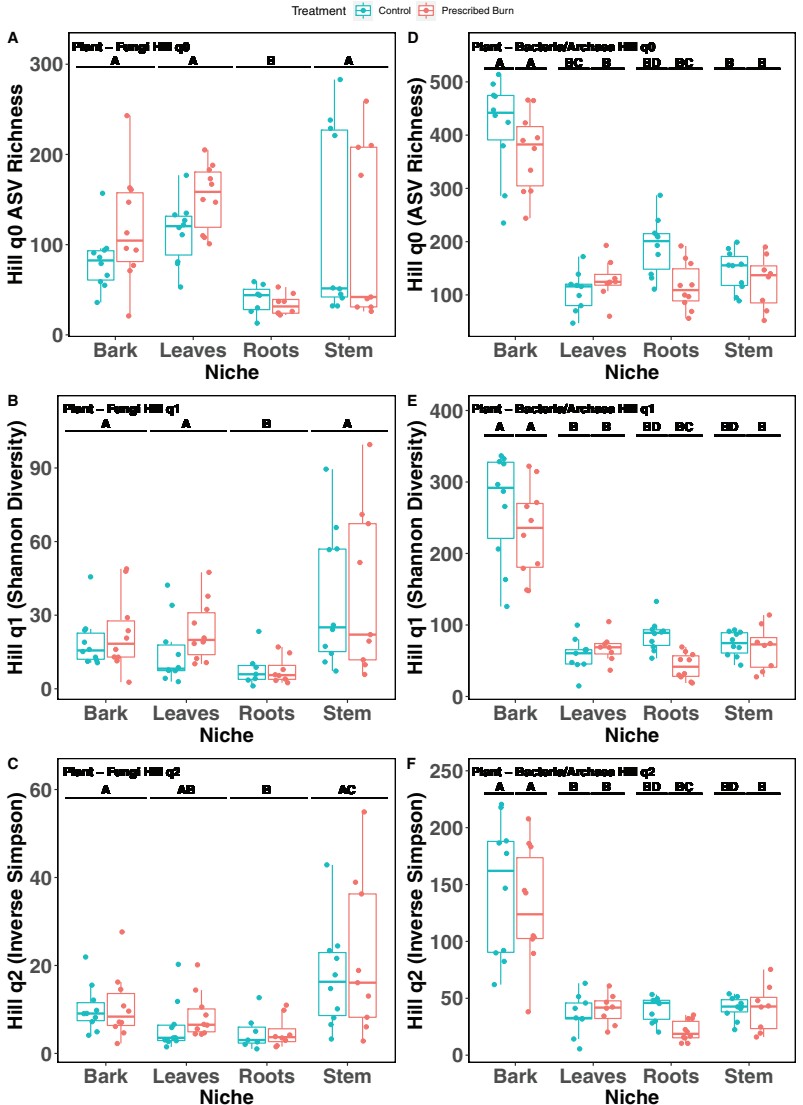

**Figure 1 Hill numbers (q0–q2) of flowering dogwood (*Cornus florida*)-associated fungal (A–C) and bacterial/archaeal (D–F) communities from unburned control and prescribed burn plots.** Letters indicate significant mean differences determined using Tukey *post-hoc* test. Color represents prescribed burn treatment.                                   

treated plots (Figs. 7B–7D). In bulk soil fungal communities, most sequences were classified to the phylum Basidiomycota, followed by the Ascomycota, Mucoromycota, and Mortierellomycota (Fig. 7E).

In flowering dogwood plant-associated bacterial/archaeal communities, the Phyla Actinobacteriota and Proteobacteria comprised the greatest number of sequences in both unburned control and prescribed burn-treated trees (Figs. 8A–8D). However, in stem and leaf-associated bacterial/archaeal communities, Actinobacteriota comprised a smaller proportion of total sequences in stem and leaf communities relative to the Proteobacteria (Figs. 8C and 8D). In root and bark communities, the proportions of these two phyla were nearly equal (Figs. 8A and 8B). In bulk soil bacterial/archaeal communities, sequences

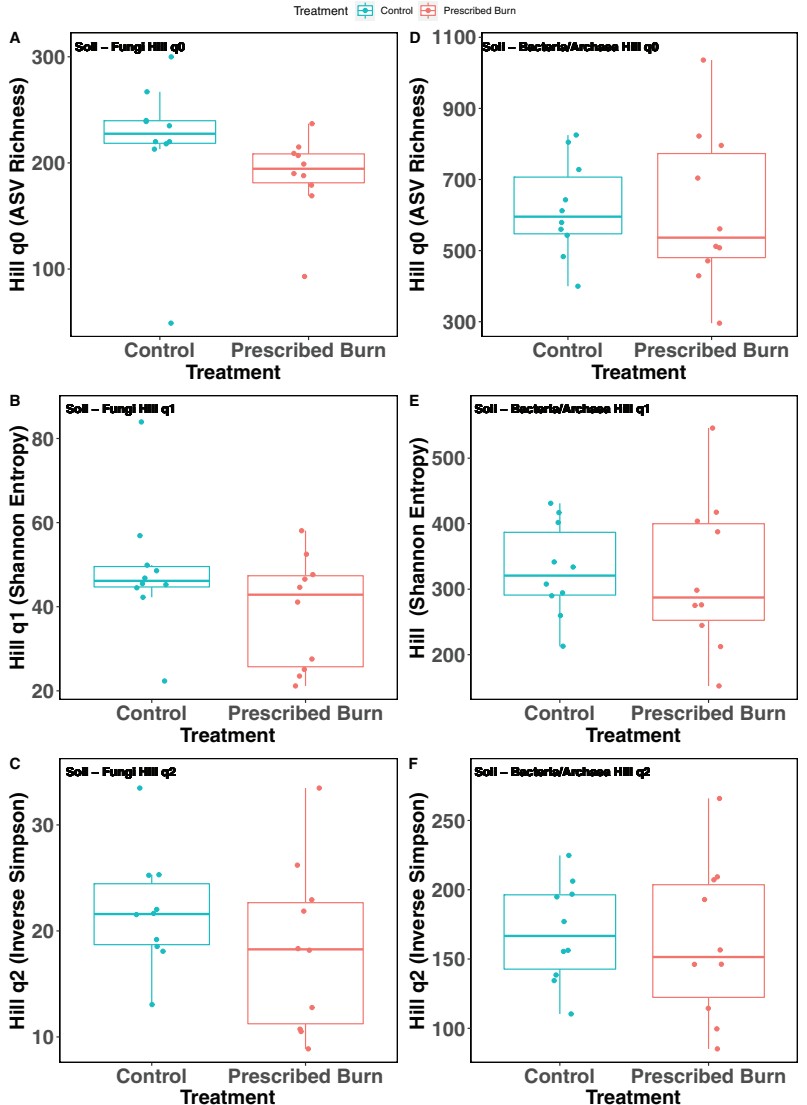

**Figure 2 Hill numbers (q0–q2) of bulk soil fungal (A–C) and bacterial/archaeal (D–F) communities of flowering dogwood (*Cornus florida*) trees.** Color represents prescribed burn treatment.

from the Acidobacteria and Proteobacteria comprised the greatest proportion of total sequences (Fig. 8E).

A total of 118 differentially abundant bacterial ASVs were detected in the roots of prescribed burn-treated trees compared to unburned control trees (Table S4). Of these, ASVs identified to the family Acidothermaceae (18/118) were the most frequently detected as differentially abundant followed by Isosphaeraceae (11/118). The relative abundances of ASVs identified to the Acidothermaceae were significantly higher in the roots of prescribed burn-treated trees compared to unburned control trees ($t = 3.47$, df = 18, $P < 0.05$ Fig. 9A). In contrast, the relative abundances of ASVs identified to the Isosphaeraceae were

**Table 4 Results of permutational multivariate analysis of variance (PERMANOVA) tests calculated for fungal and bacterial/archaeal communities of flowering dogwood (*Cornus florida*) trees.**

| | | PERMANOVA | | | | | | | | | | | |
| | | Niche | | | | Treatment | | | | Niche * Treatment | | | |
| Community | | df | Pseudo-*F* | $R^2$ | *P* | df | Pseudo-*F* | $R^2$ | *P* | df | Pseudo-*F* | $R^2$ | *P* |
| --- | --- | --- | --- | --- | --- | --- | --- | --- | --- | --- | --- | --- | --- |
| Fungi | All plant | **1.66** | **5.13** | **0.18** | **0.001** | 3.66 | 1.18 | 0.01 | 0.18 | 3.66 | 1.16 | 0.04 | 0.10 |
| | Roots | – | – | – | – | 1.13 | 1.01 | 0.07 | 0.45 | – | – | – | – |
| | Bark | – | – | – | – | 1.18 | 1.13 | 0.06 | 0.20 | – | – | – | – |
| | Stem | – | – | – | – | 1.17 | 0.72 | 0.04 | 0.81 | – | – | – | – |
| | Leaves | – | – | – | – | 1.18 | 1.91 | 0.10 | 0.07 | – | – | – | – |
| | Bulk soil | – | – | – | – | 1.18 | 1.07 | 0.06 | 0.20 | – | – | – | – |
| Bacteria/Archaea | All plant | **3.67** | **9.07** | **0.27** | **0.001** | 1.67 | 1.25 | 0.01 | 0.15 | 3.67 | 1.19 | 0.04 | 0.13 |
| | Roots | – | – | – | – | **1.17** | **1.63** | **0.08** | **0.02** | – | – | – | – |
| | Bark | – | – | – | – | 1.17 | 1.18 | 0.06 | 0.14 | – | – | – | – |
| | Stem | – | – | – | – | 1.15 | 0.89 | 0.06 | 0.40 | – | – | – | – |
| | Leaves | – | – | – | – | 1.15 | 0.94 | 0.06 | 0.50 | – | – | – | – |
| | Bulk soil | – | – | – | – | 1.18 | 1.00 | 0.05 | 0.40 | – | – | – | – |

**Note:**
Text in bold represents predictors that significantly explained differences in fungal and bacterial/archaeal community composition ($P < 0.05$).

significantly higher in the roots of unburned control trees compared to prescribed burn treated trees ($t = 3.47$, df = 18, $P < 0.05$; Fig. 9B).

# DISCUSSION

In this study, we characterized the effects of prescribed burn on the bacterial/archaeal and fungal communities of bulk soils, roots, bark, stems, and leaves of flowering dogwood trees. Contrary to our main hypothesis, significant differences by burn treatment were only detected in the alpha- and beta-diversity of root bacterial/archaeal communities. This finding does provide some support for our hypothesis that belowground microbial communities would be affected more by the prescribed burn compared to aboveground communities, potentially as a result of their proximity to the fire. In these bacterial/archaeal root communities, a significantly higher relative abundance of sequences identified as Acidothermaceae was detected. The Acidothermaceae are a family of thermophilic bacteria, lending support to our hypothesis that an increase in pyrophilic microorganisms would be observed following the prescribed burn treatment.

## Responses of abiotic and biotic soil properties to prescribed burn

The majority of examined soil physicochemical properties did not differ between the prescribed burn-treated plot and the unburned control plot. The lack of differences in soil properties could explain why we did not detect alteration of the bulk soil microbial communities following prescribed burn treatment, given the well-documented role of soil physicochemical properties in structuring microbial communities (*Parfrey, Moreau & Russell, 2018*; *Glassman, Wang & Bruns, 2017*; *Onufrak, Rúa & Hossler, 2020*; *Pulido-Chavez et al., 2021*). Most notably, SOM did not differ between the unburned

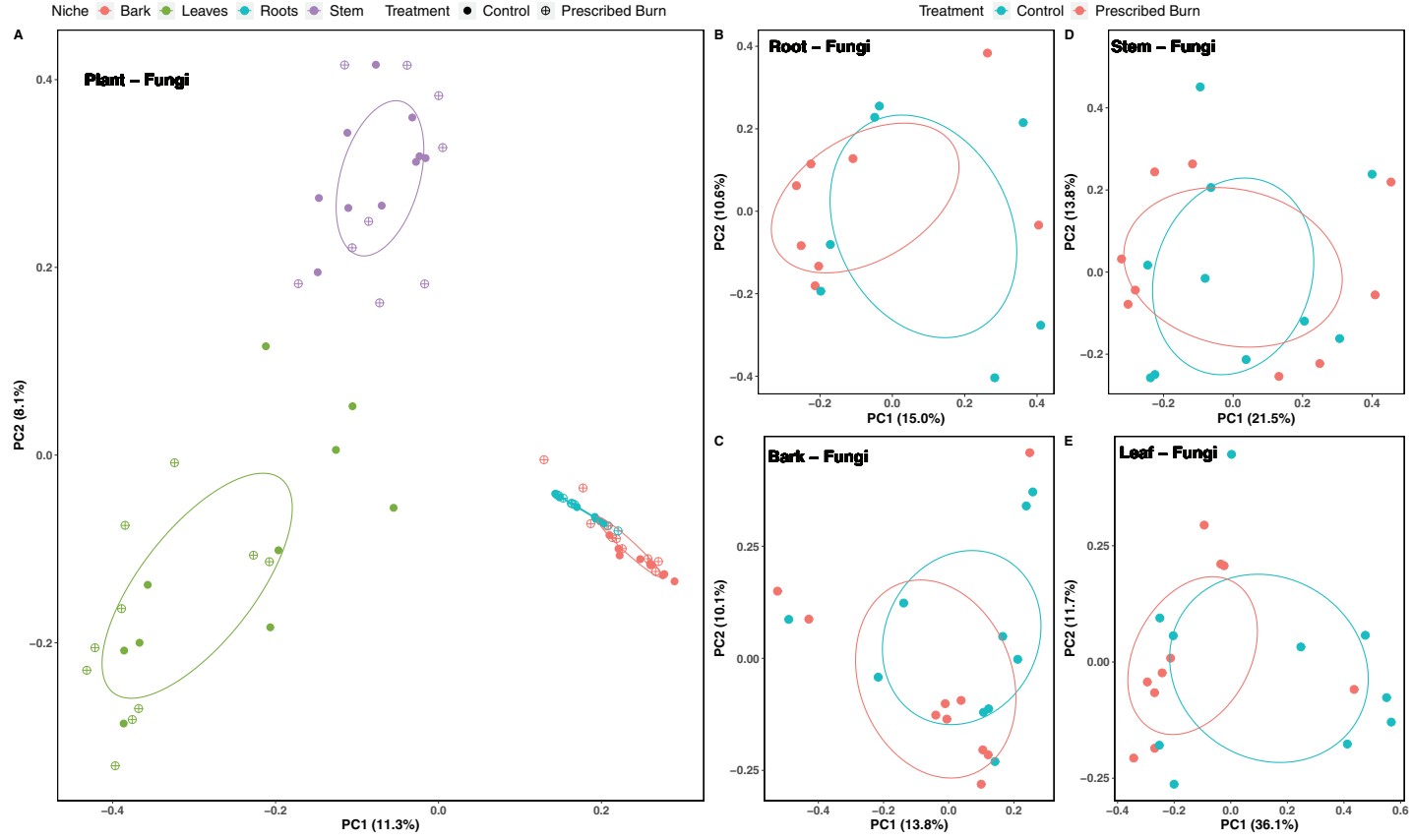

**Figure 3** **Principal component analysis (PCoA) of (A) all flowering dogwood (*Cornus florida*) associated fungal communities and (B) root, (C) bark, (D) stem, and (E) leaf fungal communities from unburned control and prescribed burn plots using Bray-Curtis distances.** In (A), color represents plant niche and shape represents prescribed burn treatment. In (B–F) color represents prescribed burn treatment. Ellipses represent standard deviation of axis scores from prescribed burn treatment centroids.

control and prescribed burn-treated plots. Previous research on the effects of wildfires on EcM fungal communities in *P. ponderosa* stands revealed that SOM was more abundant in unburned plots compared to burned plots, and that this difference in SOM was a strong driver of variation in EcM community composition (*Pulido-Chavez et al., 2021*).

In general, the effects of prescribed fire on SOM are variable and are dependent upon factors such as fire intensity, site characteristics (*e.g.*, slope, fuel abundance, and type), and weather conditions during the burn (*Mataix-Solera et al., 2011*). In addition to SOM, we did not detect significant differences in soil pH between the unburned control and prescribed burn-treated plots. Soil pH typically increases following a prescribed burn treatment (*Certini, 2005*; *Bonanomi et al., 2022*) and soil pH is a strong driver of soil microbial community composition (*Zhalnina et al., 2015*; *Parfrey, Moreau & Russell, 2018*; *Li et al., 2022*).

A lack of significant reductions in SOM and pH following the prescribed burn treatment are to be expected given that prescribed fires, particularly those applied during the cooler dormant season such as the one in our study, are lower in intensity and severity (*Fonseca et al., 2017*; *Scharenbroch et al., 2012*). While the recorded temperature of our fire exceeded

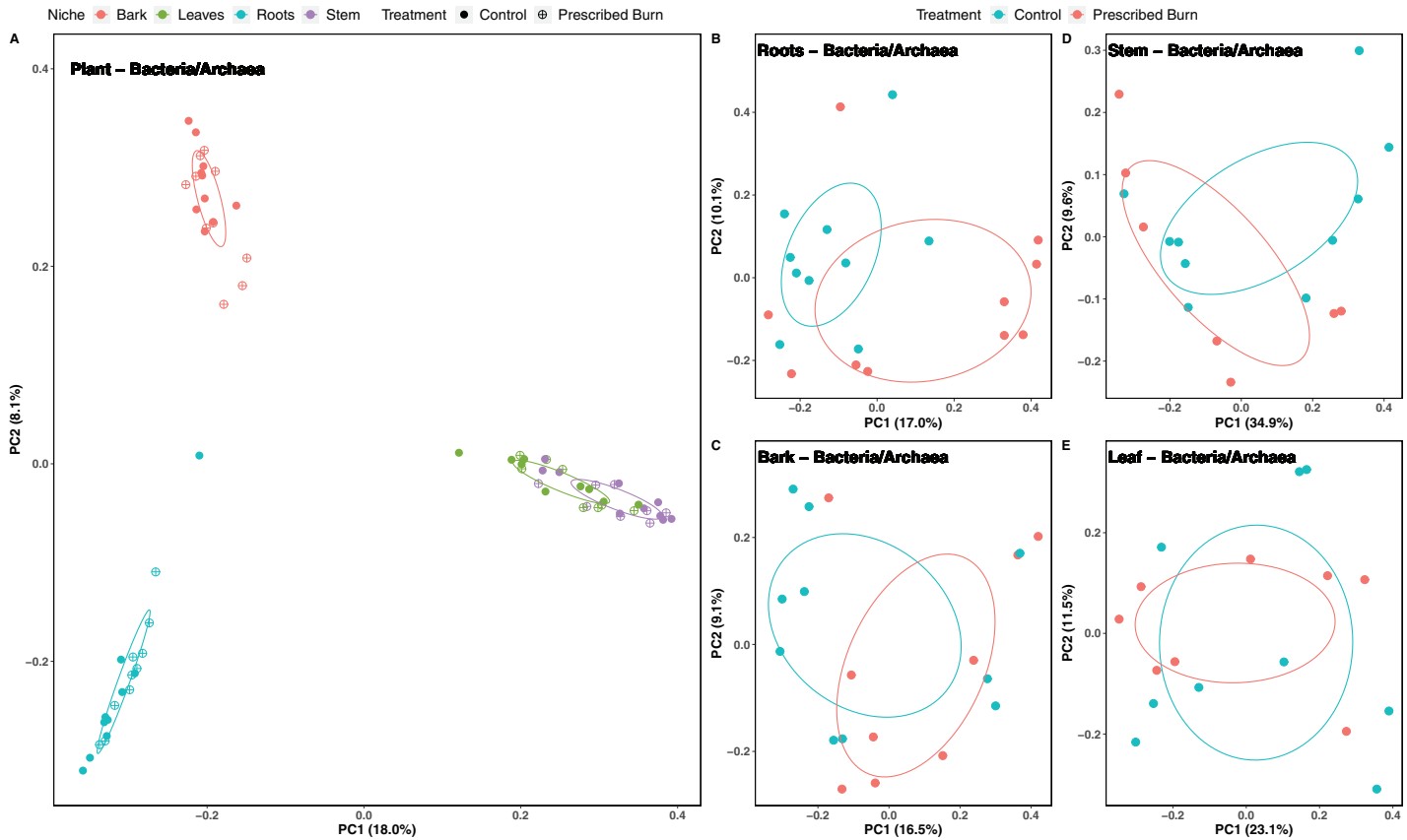

**Figure 4 Principal component analysis (PCoA) of (A) all flowering dogwood associated bacterial/archaeal communities and flowering dogwood (*Cornus florida*) (B) root, (C) bark, (D) stem, (E) and leaf bacterial/archaeal communities from unburned control and prescribed burn plots using Bray-Curtis distances.** In (A), color represents plant niche and shape represents prescribed burn treatment. In (B–F) color represents prescribed burn treatment. Ellipses represent standard deviation of axis scores from prescribed burn treatment centroids.

220 °C, which is the reported temperature at which SOM begins to combust, temperatures were recorded at the soil-air interface, which typically experiences higher temperatures than soils at lower depths (*Giovannini, Lucchesi & Giachetti, 1988*; *Mataix-Solera et al., 2011*; *Soto, Benito & Díaz-Fierros, 1991*; *Raison et al., 1986*). For instance, during a prescribed fire, the temperature of the leaf litter and soil surface was on average 600 °C and 450 °C, respectively, whereas at 2 and 5 cm beneath the leaf litter, average soils temperatures were 54 °C and 42 °C, respectively (*Raison et al., 1986*). In our study, we sampled soils to a depth of 15 cm, and as a result, may have disproportionately sampled from deeper soils that did not experience the same change in temperatures as soils closer to the surface. This could potentially dilute the overall responses of soil physicochemical properties and microbial communities to the prescribed burn treatment, hampering signal detection. There is a strong precedent for the hypothesis that bulk soil microbial community composition would shift following the prescribed fire as indicated by a meta-analysis (*Pressler, Moore & Cotrufo, 2019*) that identified reductions in both bacterial and fungal community richness in response to fire. However, *Pressler, Moore & Cotrufo*

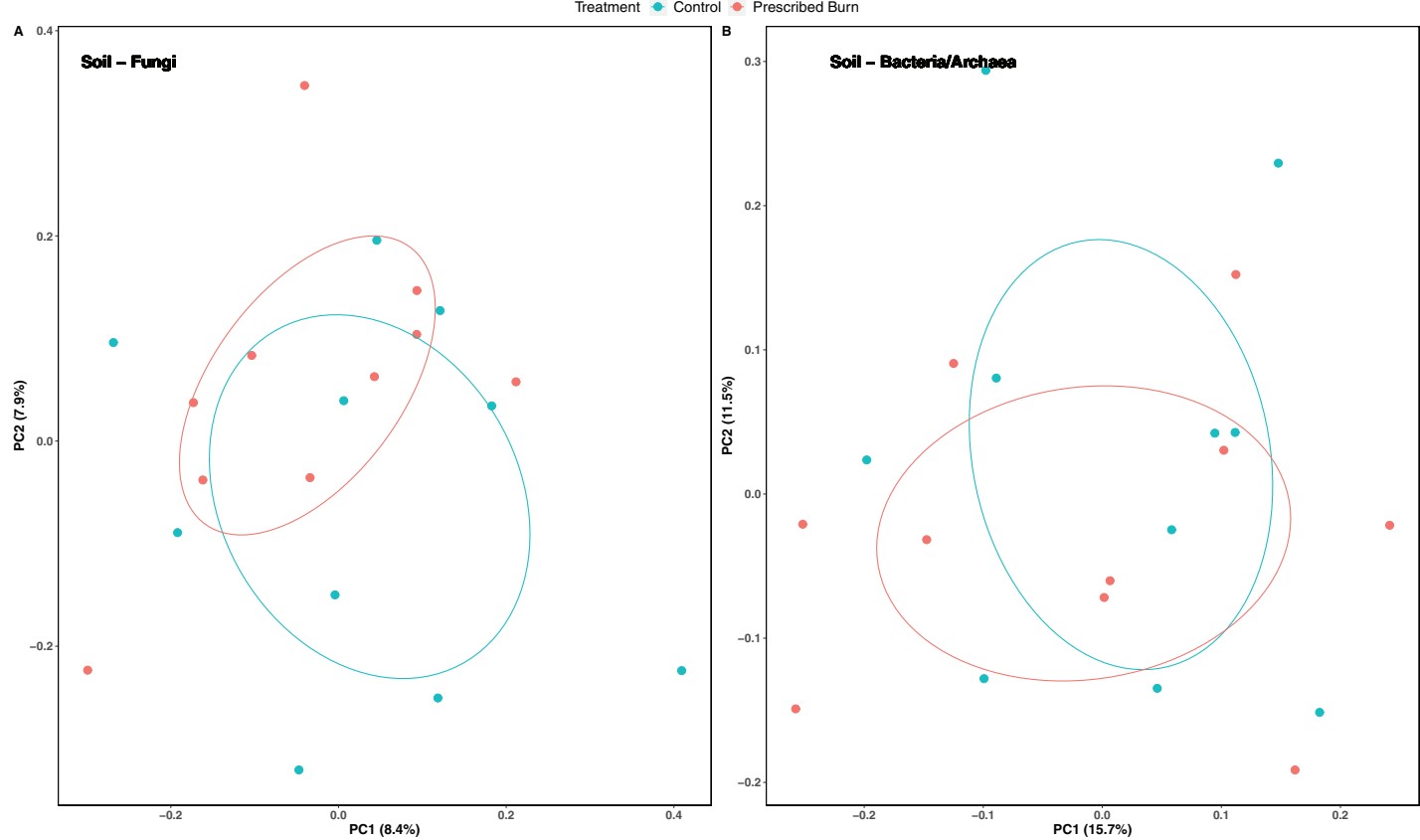

**Figure 5 Principal coordinate analysis (PCoA) of bulk soil (A) fungal communities and (B) bacterial/archaeal communities from the base of flowering dogwood (*Cornus florida*) trees in unburned control and prescribed burn plots using Bray-Curtis distances.** Colors represent pre-scribed burn treatment. Ellipses represent standard deviation of axis scores from prescribed burn treatment centroids.

*(2019)* did determine that fire type and the depth of affected soil horizons did not significantly explain changes in microbial richness and community composition, which highlights a need to further explore the mechanisms of how prescribed fires alter soil microbial communities.

In addition to soil physicochemical properties, the duration of time between burn application and sample collection may have contributed to why significant differences in bulk soil microbial communities were not detected. We collected samples approximately five months following the burn application. The duration between the burn event and sample collection significantly affects microbial community composition (*Pressler, Moore & Cotrufo, 2019*; *Pulido-Chavez et al., 2021*). As a result, our sampling date may have occurred too late following the prescribed burn for us to detect differences in bulk soil microbial communities, suggesting the effects of prescribed fire are ephemeral. However, it should be noted that *Fultz et al. (2016)* documented changes in microbial activity six-months post-prescribed burn application and *Pulido-Chavez et al. (2021)* determined that changes in EcM and saprophytic fungal communities persisted for 11 years following a wildfire. These previous findings could indicate that our sampling interval was not too long

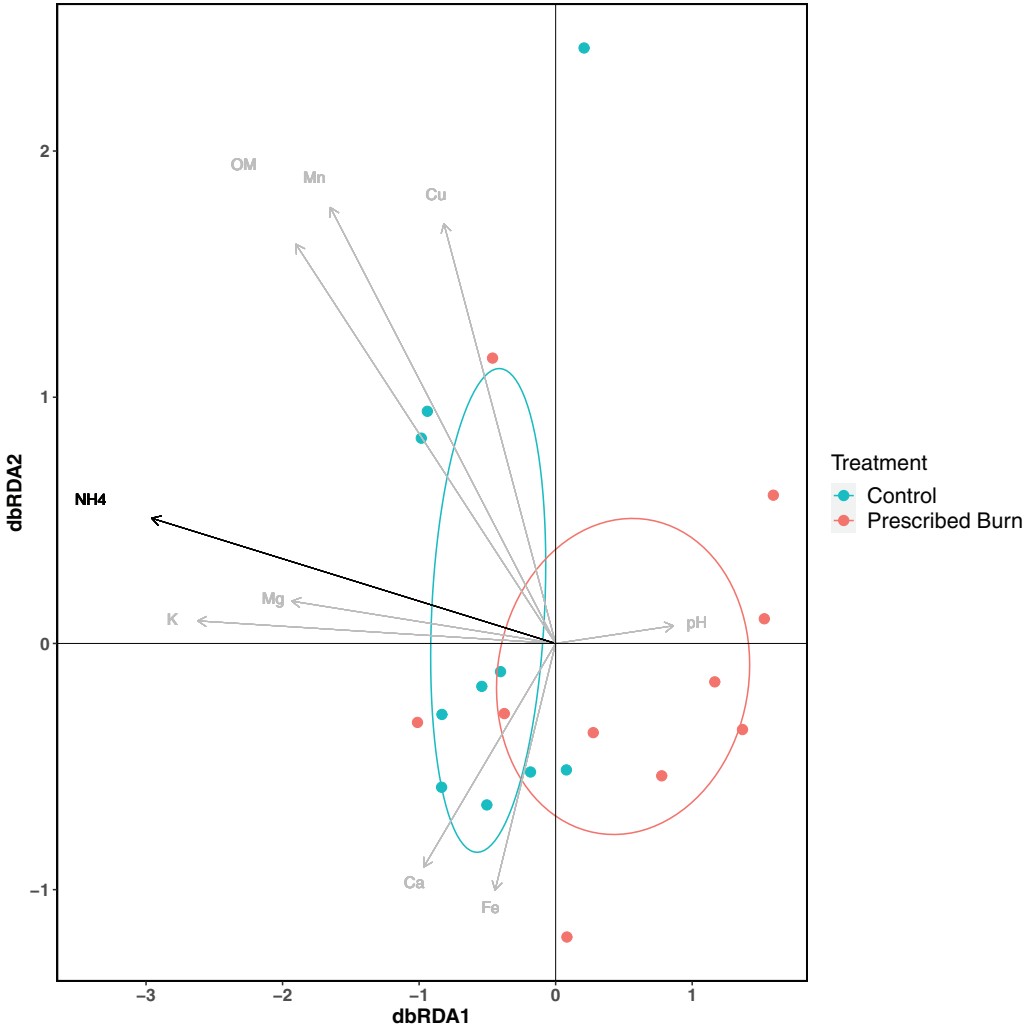

**Figure 6 Distance-based redundancy analysis (dbRDA) of soil physicochemical properties and flowering dogwood (*Cornus florida*) root bacterial/archaeal communities from unburned control and prescribed burn treated plots.** Points and ellipses are colored by prescribed burn treatment. Black arrows represent those which significantly ($P < 0.05$) drive differences in community composition whereas those in gray are not significant ($P > 0.05$). Ellipses represent standard deviation of axis scores from prescribed burn treatment centroids.

after the burn to detect a signal. Due to the possible influence of sampling interval on signal detection, future studies evaluating the effects of prescribed fires on the microbial communities of temperate forest trees should incorporate multiple sampling times following burn application.

## Responses of root microbial communities to prescribed burn

Although we did not detect differences in bulk soil microbial communities, significant differences were detected in the bacterial/archaeal communities of roots in unburned control and prescribed burn-treated plots. These differences appear to be driven in part by differences in the concentrations of soil K and $NH_4$ between the unburned control and

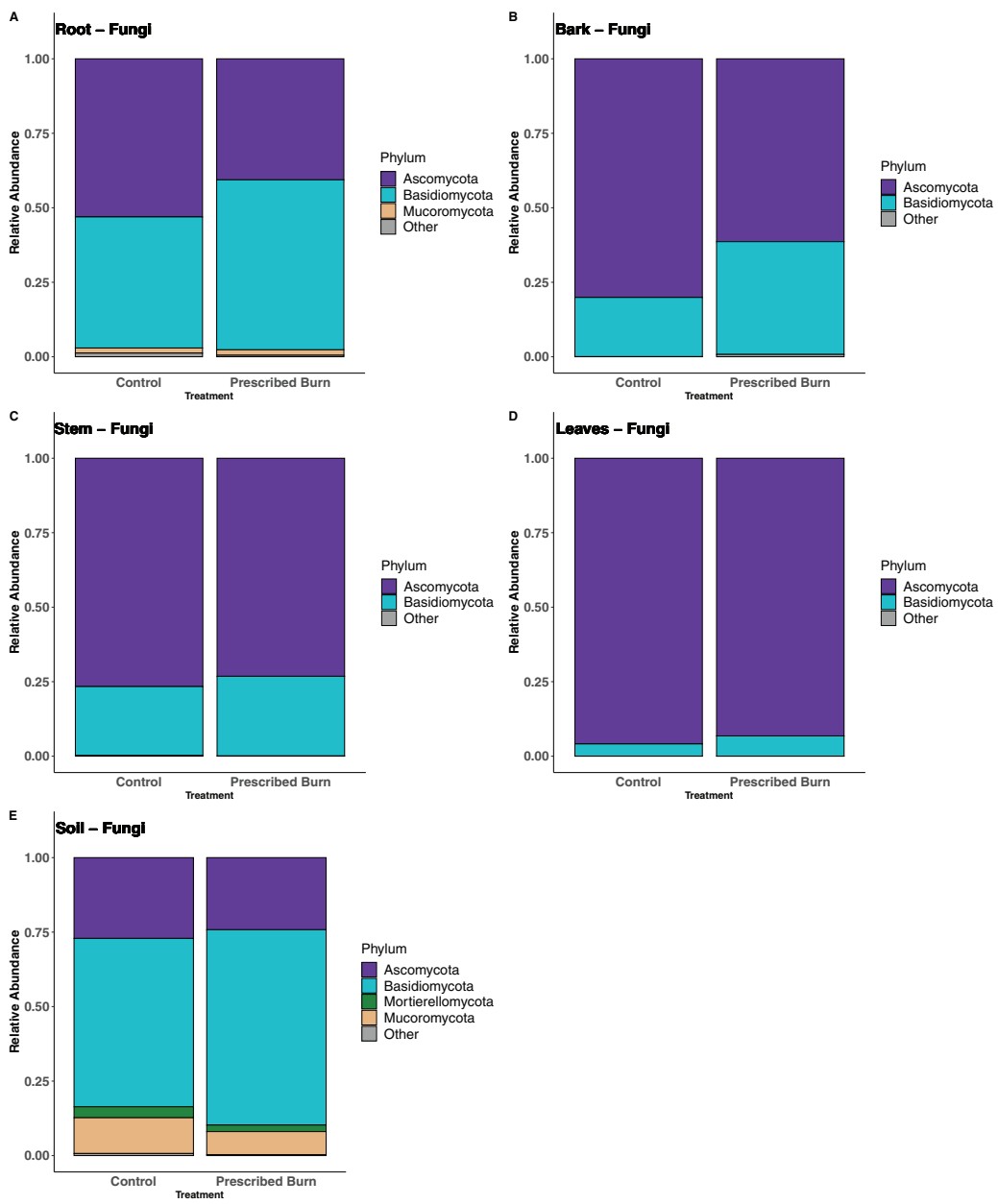

**Figure 7 Relative abundance of phyla in the fungal communities of flowering dogwood (*Cornus florida*) (A) roots, (B) bark, (C) stems, (D) leaves and (E) bulk soils.** Other category represents phyla which do not comprise at least 1% of the total sequences within each niche.

prescribed burn plots, with levels of K and $NH_4$ found to be higher in the unburned plot. Differences in soil K have been documented to affect the composition of bulk and rhizosphere soil bacterial communities (*Yu et al., 2021*), and although not significant, soil K does appear to be a strong driver of root bacterial/archaeal communities in our study. *Dove et al. (2021)* documented increases in soil $NH_4$ of prescribed burn plots compared to unburned control plots, and these changes in soil $NH_4$ were strong drivers of differences in

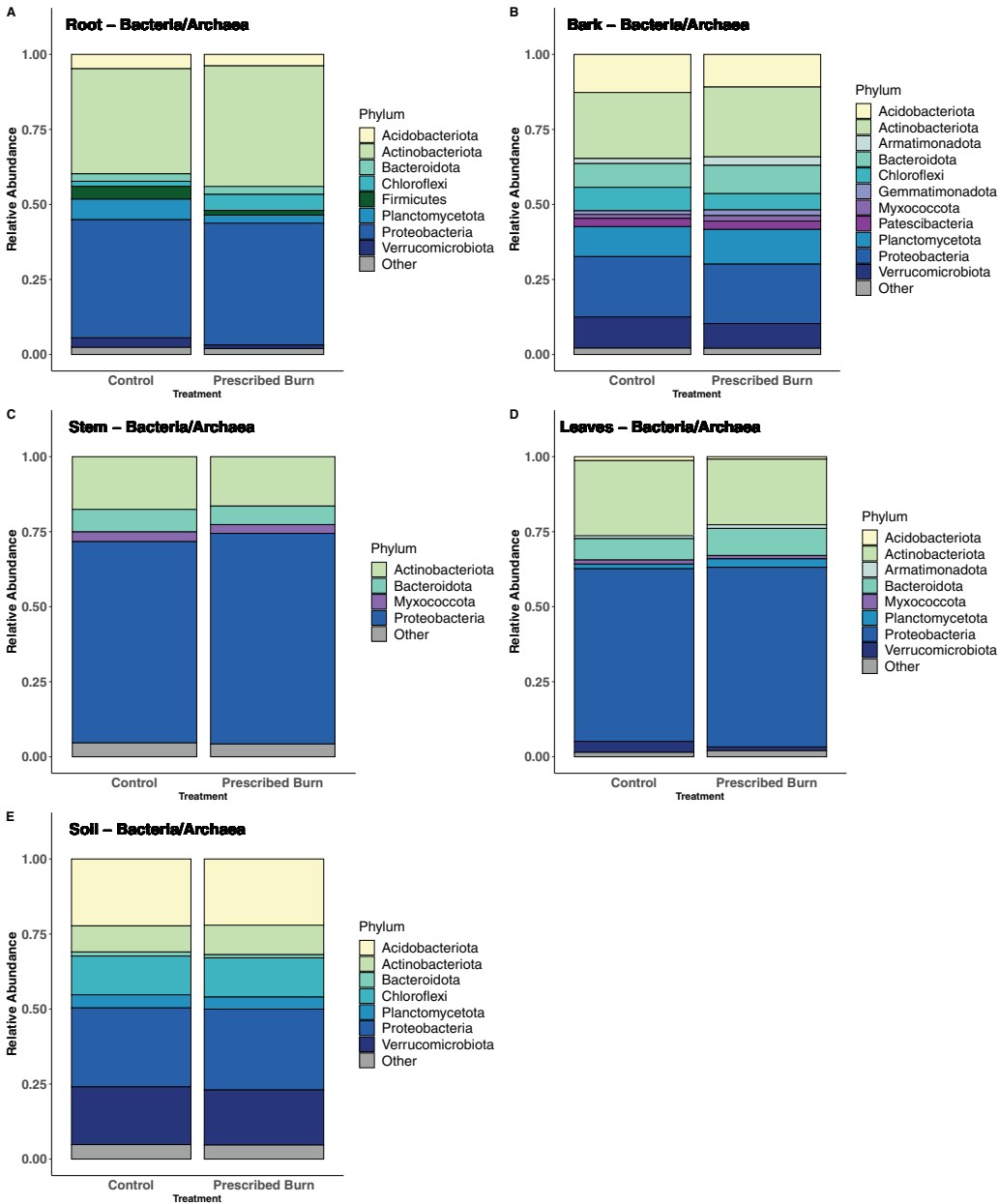

**Figure 8 Relative abundance of phyla in the bacterial/archaeal communities of flowering dogwood (*Cornus florida*) (A) roots, (B) bark, (C) stems, (D) leaves and (E) bulk soils.** Other category represents phyla which do not comprise at least 1% of the total sequences within each niche.

the bacterial/archaeal communities of rhizosphere soils. Inorganic nitrogen responses after burning vary greatly among studies (*Certini, 2005*); an effect that has largely been attributed to differences in fire severity (*Neary, Ryan & DeBano, 2005*). Over a fire severity gradient, the greatest increase in soil $NH_4$ occurred in the high severity burned plots (*Dove et al., 2021*). This provides further support for the role of fire severity in determining the response of soil $NH_4$ to prescribed fire and explains why we observed a reduction in soil

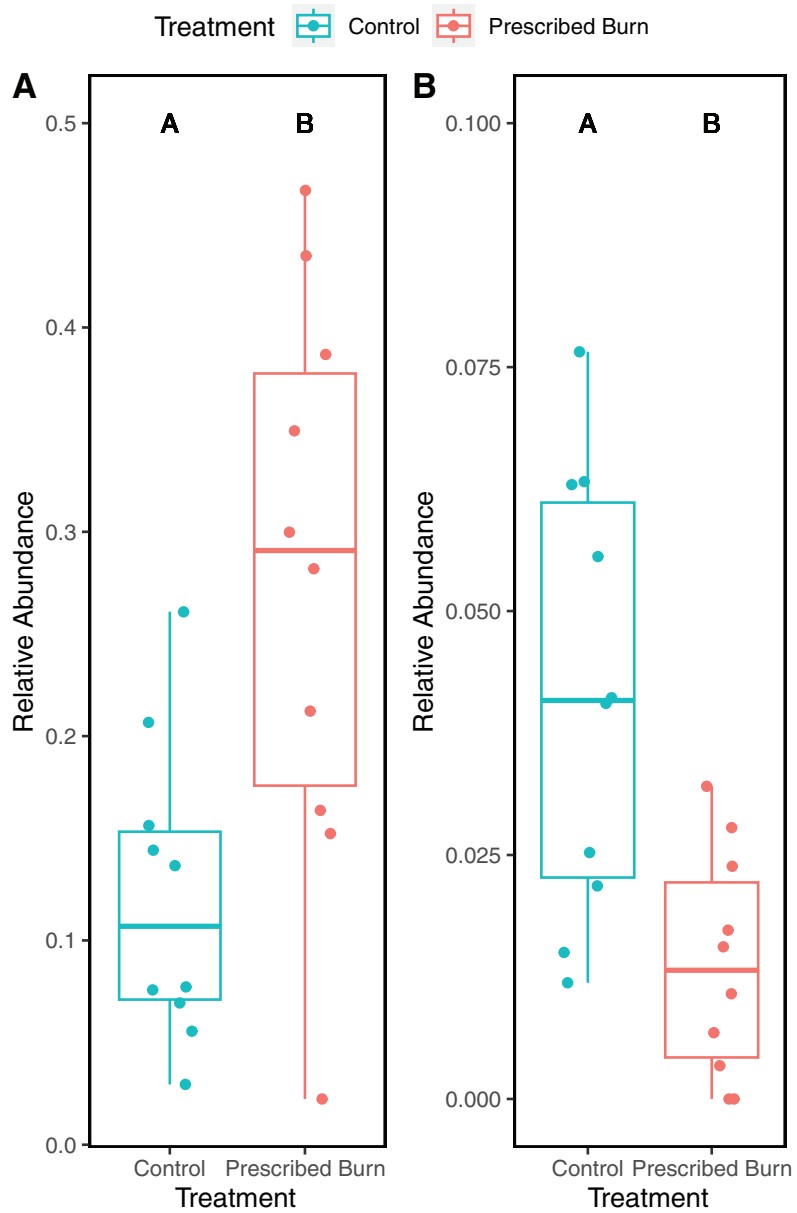

**Figure 9** Relative abundance of ASVs identified to (A) Acidothermus and (B) Isosphaeraceae in the roots of prescribed burn treated and unburned control flowering dogwood (*Cornus florida*) trees. Colors represent prescribed burn treatment.

$NH_4$ post-burn. Surprisingly, we did not detect significant differences in root fungal communities, given previous research findings showing that fungal communities are more sensitive to fire than bacterial communities (*Pressler, Moore & Cotrufo, 2019*). However, in a previous study, bacteria were more negatively impacted by fire in the short term (48 h post-fire application) compared to fungi determined using fatty acid methyl-ester profiling (*Fultz et al., 2016*). To develop a more holistic understanding of the effects of prescribed fire on microbial communities in belowground tissues, further research should focus on

changes in microbial communities across multiple time points and within plant tissues at different soil depths that occur along a fire intensity/severity gradient.

In addition to the overall differences in root bacterial/archaeal community composition, a significantly greater relative abundance of ASVs identified to the Acidothermaceae in the roots of prescribed burn-treated trees compared to unburned control trees was detected. The Acidothermaceae contains a single thermophilic bacterium that was originally isolated from thermal springs in Yellowstone National Park, WY, USA. (*Berry, Barabote & Normand, 2014*). *Acidothermus cellulolyticus* is presently the sole species in the genus *Acidothermus* of the Acidothermaceae. This thermophillic bacterial species is capable of degrading cellulose at relatively high growth temperatures (55 °C optimum) (*Mohagheghi et al., 1986*). The *Acidothermus* genome contains genes that encode for thermostable enzymatic cellulose degradation that could potentially damage the cell wall of flowering dogwood's shallow root system (*Mohagheghi et al., 1986*). The presence of these putative thermophiles five months following the burn indicates that signatures of the prescribed burn could still be detected in the bacterial communities. Additionally, ASVs belonging to the Isosphaeraceae (phylum Planctomycetota) significantly decreased in prescribed burn roots as compared to the unburned control roots. Species of the Planctomycetota are mesophilic and prefer moderate temperatures to grow (20–45 °C) (*Kaboré, Godreuil & Drancourt, 2020*). The increased relative abundance of thermophiles and reduction in the relative abundance of mesophiles suggests that sampled roots experienced high enough temperatures to impact microbial community composition, which is likely, given that roots were excavated largely at the soil surface to ensure that samples were collected from study trees.

### Responses of aboveground microbial communities to prescribed burn

We detected no significant differences in the aboveground microbial communities of prescribed burn-treated trees compared to unburned control trees. It is likely that the lack of response of aboveground microbial communities to the prescribed fire was related to their distance from the fire because stem and leaf tissues were sampled from trees approximately 5–11 m tall. In support of this hypothesis, the bark bacterial/archaeal communities of prescribed burn-treated trees tended to cluster more closely together than communities from unburned control plots. Bark was collected from the base of the tree, and most likely experienced higher temperatures than those of the canopy. Our aboveground findings were similar to *Dove et al. (2021)*, which found only a significant difference in the composition of microbial communities in the leaves of *Populus tremuloides* following a prescribed burn, but not in the stems of the trees. The response of leaf microbial communities to the prescribed burn in the *P. tremuloides* study is likely a product of the fact that the trees sampled were younger, and were found to have fungal communities largely sourced from bulk soils, which were significantly altered by the prescribed fire (*Dove et al., 2021*). The trees in our study were mature (Table S1; 29–50 years old), and as mentioned earlier, the bulk soils of our study were not significantly affected by the prescribed fire, indicating a need to understand how prescribed fire affects trees of different ages.

## CONCLUSIONS

The findings of our study suggest that prescribed fire does not significantly alter the aboveground communities of adult trees and has minimal impacts on the microbial communities of belowground tissues, at least over a period of five months. To better understand the effects of prescribed fires on plant-associated microbial communities and in turn the health of temperate forest tree species, investigations into the long-term effects of prescribed fires on bacterial and fungal communities should be further explored. This is particularly important when burn events are recurring at a location and science-based information is needed to guide appropriate forest management actions. By addressing this knowledge gap, land managers can avoid compromising the composition and function of plant-associated microbial communities and ensure that non-target effects of fire are not negatively affecting forest health (*Wardle et al., 2004*). In addition to incorporating more sampling dates and assays of targeted plant tissues made at multiple sampling depths, the inclusion of more detailed functional assays targeting either enzyme activities or genes that code for those enzymes would provide a more holistic understanding of the effects of prescribed fire on forest health. As costs of sample analyses continue to decline, study efforts could be focused on genome-wide association approaches that would better document changes within the microbial assemblage that has been affected by prescribed burn treatments (*Tabrett & Horton, 2019*).

## ACKNOWLEDGEMENTS

We thank Sarah Boggess, Anthony Moore, Michelle Odoi, Meher Ony, Grace Pietsch, and Tammy Stackhouse (University of Tennessee) for providing their expertise in DNA extraction methodologies in the laboratory.

### Funding

This work was supported by the USDA National Institute for Food and Agriculture (NIFA; Hatch project 1009630: TEN00495), the United States Department of Agriculture (USDA; Grant 58-6062-0022), and the University of Tennessee, Department of Entomology and Plant Pathology. Melissa Cregger's contribution to this work was supported by the Genomic Science Program, United States Department of Energy, Office of Science, Biological and Environmental Research, as part of the Plant Microbe Interfaces Scientific Focus Area at ORNL (Oak Ridge National Laboratory is managed by UT-Battelle, LLC, for the United States Department of Energy under contract DEAC05-00OR22725. Support for Ph.D. candidate Aaron Onufrak was provided by the USDA National Institute of Food and Agriculture AFRI Pre-Doctoral Fellowship (Grant Number: 2022-67011-36578). There was no additional external funding received for this study. The funders had no role in study design, data collection and analysis, decision to publish, or preparation of the manuscript.

## Grant Disclosures

The following grant information was disclosed by the authors:
USDA National Institute for Food and Agriculture: 1009630: TEN00495.
United States Department of Agriculture: 58-6062-0022.
University of Tennessee, Department of Entomology and Plant Pathology.
Genomic Science Program, United States Department of Energy, Office of Science, Biological and Environmental Research.
Plant Microbe Interfaces Scientific Focus Area at ORNL: DEAC05-00OR22725.
USDA National Institute of Food and Agriculture AFRI Pre-Doctoral Fellowship: 2022-67011-36578.

## Competing Interests

The authors declare that they have no competing interests.

## Author Contributions

- Beant Kapoor performed the experiments, analyzed the data, prepared figures and/or tables, authored or reviewed drafts of the article, and approved the final draft.
- Aaron Onufrak performed the experiments, analyzed the data, prepared figures and/or tables, authored or reviewed drafts of the article, and approved the final draft.
- William Klingeman III conceived and designed the experiments, performed the experiments, authored or reviewed drafts of the article, and approved the final draft.
- Jennifer M. DeBruyn conceived and designed the experiments, authored or reviewed drafts of the article, and approved the final draft.
- Melissa A. Cregger conceived and designed the experiments, authored or reviewed drafts of the article, and approved the final draft.
- Emma Willcox conceived and designed the experiments, authored or reviewed drafts of the article, and approved the final draft.
- Robert Trigiano conceived and designed the experiments, authored or reviewed drafts of the article, and approved the final draft.
- Denita Hadziabdic conceived and designed the experiments, performed the experiments, authored or reviewed drafts of the article, and approved the final draft.

## DNA Deposition

The following information was supplied regarding the deposition of DNA sequences:
    The raw amplicon sequences are available at NCBI SRA: PRJNA754133.

## Data Availability

    The codes used to analyze 16S and ITS sequence data are available at Zenodo: Beant Kapoor. (2023). beantkapoor786/Dogwood-Microbiome: dogwood_microbiome_code_v1 (Version v1). Zenodo. https://doi.org/10.5281/zenodo.7948602.

## Supplemental Information

Supplemental information for this article can be found online at http://dx.doi.org/10.7717/peerj.15822#supplemental-information.

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
