# Peer review of "Signatures of prescribed fire in the microbial communities of Cornus florida are largely undetectable five months post-fire"

_PeerJ, doi:10.7717/peerj.15822_

## Round 0.1 · original submission · Major Revisions

We have received three in-depth reviews of your manuscript. Please attend to them and carefully respond to the mentioned issues.

Reviewer 1 ·

Basic reporting

The subject of the manuscript is interesting in investigating changes in the diversity and compositions of fungal and bacterial/archaeal communities across five different plant-associated niches (soil, roots, bark, stem, and leaf) of Cornus florida L. (flowering dogwood) trees in response to a prescribed burn. However, the authors need to consider the following questions while revising their manuscript.

Abstract
Please rewrite the abstract. It is not clear that there is a significant difference in how long after the fire? The time of the study is also not specific.

Introduction
The section needs to be improved in order to better highlight the context and the novelty of the study. You also need to provide a clear guideline to understand what gaps in our knowledge you plan to fill, and you need to better introduce and justify the main objectives of your study.

Experimental design

Material and Methods
The writing has significant room for improvement. What is the planting density of trees in the sampling plot? What about climatic conditions and topographic characteristics?

Why is the burn time so designed? What is the reference basis? Please describe the details of the fire. Is it to ignite the litter? What is the original litter thickness? How high is the flame? How to put it out after 1 hour? How much litter layer burned after fire? I suggest to provide the burn scene photos.

Why did the author take samples 6 months after the fire instead of immediately? What's the point? A number of studies have reported that soils are affected by fire mostly at their very top 5 cm. Therefore, it is necessary to explain the reason why the author sampled soils from 0-15 cm.

Are the collected leaves newborn or old? I have no idea of the season when the prescribed burn was applied and the season when samples were collected.

Validity of the findings

Discussion
The section is for the time rather weak. The first subtitle should precede the second paragraph and soil physicochemical properties discussed here are mainly SOM, but it is not described in the results section. Meanwhile, the abbreviation used in the figure is OM, while the text is SOM, please unify, as well as in the Supplementary materials. In addition, the significant changes in K and NH4 are not discussed here, but are placed in the next section, which seems to be inconsistent with the subtitle.

I can't understand why the soil microorganism changes little, but the root microorganism changes significantly. Is it because the soil had recovered six months after the fire? These should be a deeper discussion.

Additional comments

Line 440: change “2021” into “(2021)”.

·

Basic reporting

Manuscript is clearly presented with sufficient references.

Experimental design

Overall fine, but with some issues - see more details below.

Validity of the findings

Data have been provided, and link between results and conclusions are clear.

Additional comments

The manuscript from Kapoor et al. describes work studying the effects of prescribed fire on microbiomes in a variety of environmental niches associated with the flowering dogwood tree. The authors report that prescribed fire impacts the root microbiome, but with minimal effects on the microbiome associated with the bark, stem, and leaves.

Overall, the manuscript is clearly presented and well written, and I appreciated the focus on different phytobiome niches that are often ignored in similar studies. The statistical analyses are appropriate for the data types and analyses being performed.

My major comment regards the bulk soil sampling and the implications for some of the resulting data analyses. The lack of signal for a microbial response was very unexpected, until it was revealed that soils were sampled up to 15 cm depth, and then homogenized. This very likely has also removed any fire-impact signal which would likely be present in the top 0-5 cm depth. I appreciate the authors acknowledging the problems with this sampling design and proposing this issue as explanation for the lack of a soil signal (L420-424). However, given the acknowledgement that any post-fire microbial signal may have been diluted by sampling across a 15-cm profile, I would also be cautious about interpretation of the soil chemistry data (e.g., L391-404), given that this would also be affected by homogenization across the same depth profile. For example, NH4 is very commonly enriched post-fire in surficial soils, and yet here NH4 is reported as being higher in the control unburned plots.

There’s a surprising lack of data on heat penetration into the soil profile during either wildfire or prescribed fire, but here are a few paper/documents that discuss this issue. They both show a pretty steep temperature gradient that could account for some of the results seen in this study:

https://www.publish.csiro.au/sr/pdf/sr9860033

http://www.secheresse.info/spip.php?article34786

Beyond this issue, I have only a number of minor comments, listed below:

L46: I would also acknowledge that historic forest management practices (and the associated build up of fuels) have also contributed significantly to the increases in wildfire size, frequency, and duration.

L54: Is there any more recent data on land in the SE US exposed to prescribed fire?

L110: It might be worth mentioning how long post-fire the burned plots were sampled

L127: Was there any prior history of prescribed burn in this area? For example, is it likely that even the control plots had been burned in the recent past?

L139: Where was that average temperature recorded? At the soil-air interface?

L160: Approximately how deep were the lateral roots sampled?

L381: Has the post-fire enrichment of Acidothermaceae been reported in other studies?

L386-389: Related to my point earlier, these lack of differences in soil properties could also be related to homogenization across the top 15-cm soil profile.

L391-293: SOM discussion – as above

L466: Given that the genus Acidothermus currently contains only one genome, I would add the proviso ‘putative’ in front of the word thermophiles


Review performed by Dr. Mike Wilkins, Associate Professor, Dept. of Soil & Crop Sciences, Colorado State University

·

Basic reporting

The aim of this work was to give a descriptive analysis of the microbiota changes following prescribed burning, the novelty stated by the authors resides in the fact that they also went for the microbiota of other plants rather than the rhizosphere which is often done. The work is interesting and much clear, and the results also are informative, stating significant differences by prescribed burn treatment only in the alpha- and beta-diversity of root bacterial communities. However, I have some concerns regarding many aspects of the work. First of all, I’d like to recommend changing the term “phytobiome” because it indicates micro and macro-organisms in plant environments not specific for microorganisms only. Moreover, many details are still missing when it comes to the burn application, like how the burn was stopped. when was it stopped? What was burnt exactly? litter quantity? What about wind speed? …
General comments:
Lines 31-32: We detected significant differences … Compared to what?
Line 37: microbial communities
Line 138: which hour of the day?
Line 154-159: It would be much better to make a subsection of chemical analysis and describe each method briefly. Also, how many replicates were used for each treatment?
Results: A table of the chemical composition of both treatments needs to be provided with the standard deviation and statistical test, and delete figure 1 instead because it creates confusion, one may think it is a biplot representing chemicals driving effects on the microbiota.
Discussion: As much as I liked the discussion, there is a big part missing regarding soil pH! There is a lot of growing evidence that prescribed burning causes a significant increase in soil pH, which is caused by the local accumulation of ash and charcoal residues (Certini, G. 2005. Effects of fire on properties of forest soils: a review. Oecologia, 143: 1-10; Bonanomi et al. 2022. Impact of prescribed burning, mowing and abandonment on a Mediterranean grassland: A 5-year multi-kingdom comparison. STOTEN, 834: 155442). And this is also evident in this work since in the PCA, we can have a high impact of pH on the ordination of microbial communities. In fact, pH is one of the strongest factors affecting soil microbiota diversity, composition, and structure.
Lines 399-401: Not true, because the highest recorded temperature as mentioned before arrived at more than 788 °C, so this could not be the reason, especially for such a 10 °C difference.
Figures:
Figure 3 should go to supplementary since all the information needed is provided in figure 2, it’s just complimentary.
Figure 6 as well should go to supplementary since figures 4 and 5 have given the necessary information.
Moreover, I don’t know why the authors have chosen to give the general differences between samples as PCoA plots, which give the general spatial distribution of the communities but don’t give details! I would like (as well as the readers) to see more details about the community composition, like stacked barplot of the microbial communities at the phylum level, heatmaps of the 100 most abundant taxa, a SIMPER test to check the main contributors of such diversity, maybe as well co-occurrence network to check how taxa are interconnected after such a burning disturbance…

Experimental design

My big concern regards the experimental design of replicates, how many replicates for each treatment were used for the chemical analysis? And in line 167, when stating “20 samples per niche”, is 20 for both treatments or for each treatment? if 10 random trees were selected for burning, how many samples for each tree were taken? did you consider all trees or sample collection was random among those 10? any composite samples? more details are needed...

Validity of the findings

No comment

---

## Round 0.2 · Minor Revisions

Please make all the minor corrections suggested by reviewers, it will read better.

Reviewer 1 ·

Basic reporting

The revised manuscript is a considerable improvement from its previous version. However, the authors need to consider the following comments.

In the abstract, the authors suggested “The alpha- and beta-diversity of root bacterial/archaeal communities differed significantly between prescribed burn and unburned control-treated trees”, whereas they concluded “prescribed burn has minimal impacts on root microbial communities of roots six months following the prescribed burn application”. The expression is unclear and even contradictory.

Line 54: Why give special examples of “xeric forests” and what is their relationship to this study? The authors need to elaborate in the text.

Experimental design

Line 97: How to understand “cooler prescribed fire”? Compared to wildfires? Whether it's really cooler or not depends on many factors. I don't think this is a good expression. I almost misunderstood that the author designed a “cooler prescribed fire”.

Line 131: This paragraph should provide additional information on local climatic conditions. We are not sure if March is in spring or late winter, although the author noted autumn six months later. Moreover, slope and other topographical information have an impact on the prescribed fire.

Line 152: The authors need to explain that the fire naturally extinguishes without human interference, and the amount of litter after the fire is undoubtedly reduced. This expression is meaningless. What we are more interested in is how much litter has decreased.

Line 162-163: I don't understand why the authors equate 152 days to about 6 months.

Validity of the findings

Lines 124-126: Is the prescribed fire a surface fire rather than an underground fire? By comparison, shouldn't it be that the bark has more contact with fire than the root? The theoretical support for the author's hypothesis seems to be weak.

·

Basic reporting

Manuscript is clear and well presented

Experimental design

The additional details surrounding the experimental design have now strengthened the paper

Validity of the findings

Statistics are appropriate and the data and supporting code is available.

I believe that the results have been interpreted appropriately.

Additional comments

Just a few minor things that should be modified prior to publication:

1. L78 - (phytobiome hereafter) needs to be removed

2. Starting L639, the authors state that bacteria are more significantly affected by fire than fungi. I'm not sure that it's as clear as this - there are plenty of studies stating the opposite, including the meta-analysis from Pressler that is cited earlier in the manuscript. I suggest that this statement be refined to reflect some nuance.

·

Basic reporting

Satisfied with the author's responses to my concerns

Experimental design

Clear

Validity of the findings

Valid

---

## Round 0.3 · accepted · Accept

The manuscript is now ready for publication, congratulations!!